# The relationship between digital media use during pregnancy, maternal psychological wellbeing, and maternal-fetal attachment

**Melissa Smith[1], Annaleise S. Mitchell[1,2], Michelle L. Townsend[1,3], Jane S. Herbert[1,2]***

**1** School of Psychology, University of Wollongong, Wollongong, New South Wales, Australia, **2** Early Start, University of Wollongong, Wollongong, New South Wales, Australia, **3** Illawarra Health and Medical Research Institute, University of Wollongong, Wollongong, Australia

\* herbertj@uow.edu.au

**Data Availability Statement:** All relevant data are within the manuscript and its Supporting information files.

## Abstract

The widespread accessibility and use of the internet provides numerous opportunities for women to independently seek out pregnancy-related information and social and emotional support during the antenatal period. Given the heightened psychological vulnerability of the pregnancy period there is a critical need to examine digital media use within the context of the feelings that women have about themselves and towards their fetus. The current study examined the relationship between digital media use during pregnancy, psychological well-being and their maternal-fetal attachment using an online survey. Forty-eight pregnant women completed a self-report questionnaire on their reasons for using digital media, and standardised measures of self-criticism, negative affect, social quality of life (QOL), and maternal-fetal attachment. The mean age of participants was 29.4 years (*SD* = 5.26), with a mean of 24.3 weeks gestation (*SD* = 9.95). Information seeking, emotional support and social support were highly endorsed reasons for digital media use (85.42%, 66.67%, 62.5% respectively). However, digital media use was positively correlated with negative affect (*p* = .003) and self-criticism (*p* < .001). Digital media use was also negatively correlated with QOL (*p* = .007). There was no evidence of a relationship between digital media use and maternal-fetal attachment (*p* = .330). Digital environments may be an important social context within which a pregnant woman develops her own maternal identity and knowledge. There are a number of benefits and limitations of this medium for providing information and support for women during pregnancy. Enhancing the opportunities to promote pregnant women's wellbeing in this context is an important avenue for further research and practice.

## Introduction

Pregnancy represents a time of transition that can be rewarding and challenging, both physiologically and psychologically [1]. Although pregnancy is regarded as a positive time by many women, common sources of reported distress include loss of sense of self, changing physical appearance and comparison to other women, concerns about not bonding with their baby, and the possible impact of the distress itself on their developing fetus [2]. A significant number of women experience mental health issues such as anxiety and depression in the antenatal or

**Funding:** Data analysis and the preparation of this manuscript was supported by Discovery Project funding (DP180101286) from the Australian Research Council (arc.gov.au) awarded to JSH.

**Competing interests:** The authors have declared that no competing interests exist.

postnatal periods [3, 4]. Changes in maternal wellbeing occur across pregnancy and postpartum, with high maternal distress particularly prevalent during the third trimester compared to 12 weeks postpartum, and significant decreases in health related Quality of Life (QOL) across this transition [5].

The social context within which the positive and negative experiences of pregnancy are occurring extends well beyond in-person interactions. Digital media comprises of many forms of digital technology that allow for the dispersal of information including; informative websites and applications on a smart phone [6]. A large component of digital media is social media, which refers to internet-based channels of mass communication enabling interactions among users, with the content being primarily user-generated [7]. In the general population, digital media use (e.g., the internet) is widespread. An estimated 90% of Australians report that they are on social media, with adults spending an average of 2.6 hours per day using social media [8]. Similar figures are reported in the USA [9] and UK [10]. Within this context, pregnant women are increasingly using the internet to access social and emotional support [11–13], and as a source of information on pregnancy-related topics such as nutrition [14]. Although only one study [15] has examined the impact of technology on maternal mental health in pregnancy, finding limited association, the authors argue the importance of further investigating the type of technological activities undertaken to understand how they facilitate social support and wellbeing. Given that pregnancy is recognised as a time of heightened psychological vulnerability [16], and the limitless opportunities for accessing information and support from online sources, there is a critical need to better understand the relationship between digital media use and the wellbeing of women during this important life transition.

Social support plays an important role in maintaining psychological wellbeing during this time. Akiki et al. [17] found that women who receive greater social support from their family reported feeling significantly less anxious than those who did not. Pregnant women and new mothers have reported that they use and value digital media for providing access to emotional and social support from their family, friends, and other new mothers [11, 18], as well as an immediate source of information [12]. By connecting with other expecting and new mothers through online forums, there is the opportunity to discuss topics that a woman may not feel comfortable sharing with their family and friends or with others in face-to-face settings [19, 20]. In individual interviews conducted by Johnson [20], women who were becoming mothers for the first time reported gaining knowledge and support from reading other women's stories in online mothering spaces. Although some women reported the usefulness of actively participating in online social interaction, by asking questions or commenting on other women's posts, the majority of women reported benefiting from these online spaces while remaining anonymous and engaging only as an observer. In focus groups, women have also reported using apps and websites to obtain regular information (such as about stages of pregnancy), intimate information considered too private to discuss with friends or family, practical information on parenting, and to gain reassurance and support when worries arise [11, 12]. Research has also found that pregnant women are using the internet to skilfully navigate information and avoid anxiety caused through information-seeking [21], to find information independently and to gain additional information to that provided by their healthcare professional [14, 22].

While internet use appears to help meet the "information need" of women during pregnancy [22–24], online searches and behaviours that might be driven by a women's need for reassurance about herself or her pregnancy could reinforce anxiety. For example, if a woman places high expectations on herself about how she should be feeling and what she should be feeling during pregnancy, it can lead to feelings of guilt and self-criticism [2]. Priel and Besser [24] found that measures of self-criticism were negatively correlated with social support in first-time mothers, with the authors suggesting that women with high self-criticism may see

social support as limiting one's autonomy and reinforcing their perceived inadequacy as a mother. Similarly, Felder et al. [25] found severity of depression and anxiety scores to be positively associated with self-judgement and isolation, and negatively associated with self-kindness. Increased self-criticism and decreased self-compassion have also been identified as risk factors for developing depression [26]. Given that women report using online forums, apps, and websites to gain reassurance and support when worries arise during pregnancy and early motherhood [11, 12], and to read other women's stories [20], it is important to better understand the extent to which digital media use may interact with feelings of self-worth and self-criticism.

In addition to the psychological wellbeing of the woman as an individual, an emotional connection may also begin to develop during pregnancy from the woman towards her fetus. This emotional bond has been referred to as Maternal-Fetal Attachment (MFA) [27]. Although there is some contention over the best way to define this bond in terms of the traditional attachment relationship [3], MFA broadly refers to the mother's attempts to love, care for, and protect her unborn child. It is during this process that a woman develops her identity as a new mother, develops an identity for her fetus, and learns about the relationship between herself and her fetus [28]. The extent to which women engage in behaviors that represent a relationship with their unborn child forms the basis of the child's socialization process. When attachment figures do not exhibit nurturing and protective behaviors towards their infants, the infant's social, emotional and cognitive development may be impaired [29, 30]. This same consequence can be seen if these behaviors are not observed during pregnancy, highlighting the importance of enhancing MFA [31, 32].

Although limited research has examined links between MFA and digital media use, initial studies suggest that digital technology which provides opportunities for social and emotional support may be related to MFA, in the same way that traditional social support has been found to be related to MFA [33]. Using semi-structured interviews and focus groups, Ross [34] found that the interviewees enhanced their MFA through technologies that allow visualization of the developing fetus. Another study focused on the effect of ultrasounds on MFA, and found that there was a significant difference in MFA from before the scan to two weeks after [35]. These studies suggest that being able to visualize one's fetus, either through an ultrasound or through digital media, may be associated with higher levels of MFA. Recent research suggests that the use of smartphone apps may increase a mother's capacity for 'mind-mindedness', which may predict secure infant-caregiver attachment [36], indicating that the use of digital media may have positive consequences for the mother, child, and their developing relationship.

The literature reviewed above confirms the significant role that internet searches, online discussion forums, and apps, play in supporting the health information seeking and decision making of women during pregnancy [11, 12, 14, 21–23]. However, possible relationships between digital media use, women's psychological wellbeing, and the developing maternal-fetal relationship during pregnancy have yet to be examined. The aim of the current study is to explore the relationship between social, emotional and informational support through digital media and a woman's psychological wellbeing, including negative affect, social Quality of Life (QOL) and self-criticism, and MFA during pregnancy.

## Methods

### Participants

Participants were women recruited during the antenatal period. Inclusion criteria was being pregnant and having a proficient understanding of English to complete the surveys. The study was open to women of all gestational ages. For the purposes of recruitment, a Facebook post

describing the study was generated. This included a brief description and a direct link to study consent information and survey. The recruitment post was posted on various mothering, pregnancy support and community groups on Facebook Australia-wide. Further recruitment was facilitated by the snowballing method. Recruitment occurred across a 3 month period, from May to July, 2018. Attrition was greatest at the beginning of the study, with 140 individuals indicating their consent to participate, but only about half of these ($n$ = 72) completing the first questionnaire. To be included in these analyses, participants were required to complete the entire questionnaire package (final sample $n$ = 48 described below). This research was approved by the University of Wollongong Human Research Ethics Committee (approval number 2018/162).

## Measures

**Digital media questionnaire.**   A Digital media questionnaire (see S1 Appendix) was developed for the purposes of this study. The questionnaire contained 13 preliminary questions, and 11 questions that were used to determine a Digital Media Use Score. Questions were informed by the themes of information seeking and providing reassurance that have been identified in previous research conducted with Australian women examining experiences of using digital media for pregnancy and parenting purposes [11–13, 37]. Preliminary survey questions asked participants to select the forms of digital media they used during this pregnancy, to order the relative importance of online and in person sources of pregnancy advice and resources, and to select their main reasons for digital media use. Follow-up questions on information seeking addressed timing of use, including in relation to doctor/midwife appointment, and reasons for doing this. For two questions, the opportunity for a free text explanation was provided if a yes response was given: "Do you use digital media to find information regarding the development of your fetus during this pregnancy?" and "Do you use digital media to find information regarding your health during this pregnancy?". The frequency of digital media use for information seeking, and for social and emotional support was determined through six-point Likert scales (more than five times a day, two to five times a day, once a day, three to five times a week, one to two times a week and less than once a week). Three additional questions which focused more specifically the individual's frequency of updating of social networking sites and perceptions on the credibility of social media profiles were not included within the current study analysis.

The final section of the questionnaire presented 11 statements which focused on the individual's cognitions, subjective experiences and social comparisons made on digital media. Example items included, "I use digital media to interact with other parents-to-be". Response options were given for these questions on a five-point Likert scale from (1 "never" to 5 "always"). Responses scores for these statements were summed to provide a Digital Media Use score, with a higher score indicating more frequent usage of digital media for pregnancy-related information and support. Reliability analysis for these 11 statements indicated acceptable internal consistency $\alpha$ = .95.

**Depression, anxiety and stress scale-21 (DASS-21).**   The DASS-21 is a 21-item self-report questionnaire that assesses psychological distress experienced over the past week [38]. The DASS-21 contains three subscales: depression, anxiety and stress. Respondents rate on a scale from 0 to 3 (never, sometimes, often, almost always) the extent each statement has applied to them over the past week. Examples of statements include: "I found it hard to wind down" and "I felt that I was using a lot of nervous energy." As we were interested in using the measure as a broad indicator of negative affect, we generated a mean score rather than analysing each domain separately [39]. The range of these mean scores were 0 to 42, with higher scores

indicative of higher levels of negative affect. The DASS-21 shows high reliability and internal consistency [40] and exhibited a high level of internal consistency in the current study, $\alpha$ = .95.

**Depressive Experiences Self-Critical Subscale-6 (DEQ-SC-6).** The DEQ-SC-6 [41] is a six-item measure of trait self-criticism derived from the original Depressive Experiences Questionnaire [42]. Participants responded to statements on a seven-point Likert scale (1 "strongly disagree" to 7 "strongly agree"). The scores were summed and higher scores were indicative of higher levels of self-criticism. Examples of statements include "I find it hard to accept my weaknesses" and "I compare myself often to standards or goals". Reliability analysis indicated acceptable internal consistency $\alpha$ = .86.

**The World Health Organisation Quality of Life Scale (WHOQOL-Brief).** The WHO-QOL-Brief was developed by the World Health Organisation (WHO) [43] as a shortened version of the WHOQOL-100, measuring quality of life, applicable across cultures. The WHO-QOL-Brief contains 26 questions across four domains: physical, psychological, social relationships and environment. The items are scored on a five-point Likert scale, with mean scores derived for each domain and multiplied by four to be comparable against scores in the WHO-QOL-100. As we were interested in social quality of life, only this domain was interpreted in the present study. Higher scores are indicative of higher social quality of life. The WHOQOL-Brief has good reliability and internal consistency [44], and exhibited a high level of internal consistency in the current study, $\alpha$ = .93.

**Maternal-Fetal Attachment Scale (MFAS).** The MFAS is a reliable and valid 24-item scale developed by Cranley [28] to measure maternal-fetal attachment. One item that asks participants about self-harm behaviour was removed from the present study, due to the research not being conducted in a clinical setting and researchers being unable to provide appropriate action to participants' responses. Higher scores on the MFAS were indicative of higher MFA. Reliability analysis indicated acceptable internal consistency $\alpha$ = .89.

**Demographics form.** The demographics form was comprised of 36 questions developed for the current study. It consisted of questions regarding personal demographics (age, country of birth, marital status, and education levels) and pregnancy-related questions (current gestational age, current and previous pregnancy history).

## Procedure

Participants were invited to complete the anonymous survey online via the platform Survey Monkey. After clicking on the invite link, participants were presented with the participant information sheet followed by the consent form. Participants gave their informed consent by checking the "I agree" box on the bottom of the consent form. Participants could withdraw at any time by exiting the survey. No questions were compulsory to answer, and the questionnaire was voluntary and non-commercial. No incentive was given for participation.

## Data analysis

Data was analysed using the Statistical Package for Social Sciences (SPSS). Pearson correlations and Spearman's Rho were used to assess the relationship between overall digital media usage from the 11-Likert scale questions and each of the measures of psychological wellbeing (DASS-21, DEQ-SC-6 and social QOL), as well as the age of participants. Qualitative responses from the digital media survey were used to assess the reasons behind digital media usage during pregnancy.

Responses to open text questions were analysed using thematic analysis. This study adopted Braun and Clarke's [45] procedure for using thematic analysis to analysis qualitative data within psychology. Provisional codes were identified by the first author (MS) to capture participant

comments that were noteworthy and which featured regularly. These were checked by the second author (AM) and categorised into themes which were then reviewed by the research team to ensure they were reflective of the data's narrative and research aims.

One-way between groups analysis of variance was used to compare the difference in MFAS scores between participants who used digital media for social and emotional support either less than once a week, between once and five times a week, and between once to more than five times a day. One item in the stress scale of the DASS-21 was written incorrectly, so answers to this item were treated as missing data. The mean score of the other six items of the stress scale were used in replacement for the missing data to enable correct scoring and scale use [39].

# Results

## Participant characteristics

Participants ($N = 48$) were between 19 and 43 years of age ($M = 29.4$, $SD = 5.26$), with the majority indicating that they were born in Australia. The remaining countries of birth were USA ($n = 7$), New Zealand, UK, Poland and Vietnam (all $n = 1$). A wide range of gestational ages were reported, ranging from 2 weeks to 39 weeks, with a mean of 24.3 weeks gestation ($SD = 9.95$). Table 1 outlines the demographic information of the participants.

Table 2 presents a summary of mean, standard deviation and range of scores for outcome measures.

## Digital media use during pregnancy

The majority (85.42%) of participants endorsed information seeking as a reason for using digital media during their pregnancy. Social support (66.67%), emotional support (62.5%), and sharing photos (43.75%) were also endorsed as common reasons for using digital media. Only one participant indicated that none of the listed reasons were why they used digital media, and

**Table 1. Participant demographics.**

|  |  | Participants ($n = 48$) |
|---|---|---|
| Mean age in years (SD) |  | 29.4 (5.26) |
| Place of birth | Australia | 72.9% |
| Highest qualification |  |  |
|  | University Degree | 47.9% |
|  | Vocational Qualification | 22.9% |
|  | Completion High School | 10.4% |
|  | Completing Year 10 or less | 14.6% |
| Married |  | 64.6% |
| Pregnancies |  |  |
|  | Primipara | 46% |
|  | Multipara | 54% |
| Gestation in weeks (SD) |  | 24.3(9.95) |
| Planned pregnancy |  | 72.9% |
| Pregnancy from fertility treatment |  | 8.3% |
| Previous miscarriage |  | 27.0% |
| Employment | Full time | 41.6% |
|  | Part time | 27.0% |
|  | Unemployed | 25.0% |
|  | On leave | 6.3% |

**Table 2. Mean (SD) and range of scores on key wellbeing outcomes.**

| Measure | Mean (SD) | Range |
|---|---|---|
| Maternal-Fetal Attachment (MFA) | 88.8 (12.33) | 54–112 |
| Total DASS-21 | 8.7 (8.53) | 0–53 |
| Social domain QOL | 68.4 (20.55) | 17.67–100 |
| Self-criticism (DEQ-SC-6) | 21.9 (8.58) | 6–42 |

provided their own response of "having down time". The mean digital media use score was 34.7 ($SD$ = 7.04; ranging 14 to 51), indicating that these participants were frequent users of digital media for obtaining pregnancy related information and support. As shown in Fig 1, a significant negative bivariate correlation was found between digital media use and age, $r(46)$ = -.34, $p$ = .018, with younger users more frequently using digital media for pregnancy information and support than older users.

In response to the question, "Do you use digital media to find information regarding your health during this pregnancy?" the following themes were identified: determining what is normal, and information and reassurance seeking (e.g. food safety, common symptoms and relief, nutrition and exercise). Table 3 presents example quotes for each theme. Eight participants (16.67%) indicated that they did not use digital media for information seeking regarding their health.

Table 4 presents common themes regarding why participants engaged in this type of information seeking, and the percentage of participants that endorsed each theme. The majority of participants indicated that they used digital media to find information regarding the development of their fetus, with only $n$ = 6 (12.5%) participants indicating they did not use digital media for this purpose. A small portion of participants ($n$ = 6, 12.5%) indicated that they were not likely to use digital media for information seeking before an antenatal appointment. For those likely to use digital media before an appointment, the most endorsed reasons for doing so were to prepare for questions ($n$ = 33, 68.75%), to ease nerves ($n$ = 33, 68.75%), and to feel knowledgeable ($n$ = 32, 66.67%). Only two participants indicated 'other' reasons, and specified these as "reviews on doctor" and "just to get a general idea"

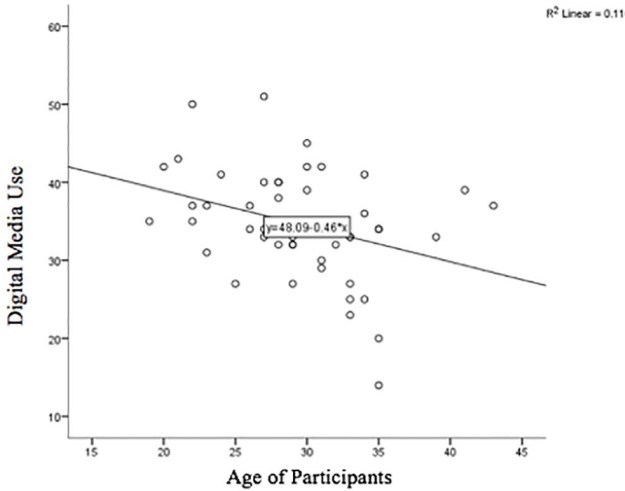

**Fig 1. Linear relationship between overall digital media usage for information and support during pregnancy and the age of participants.**

**Table 3. Indicative quotes and their respective themes regarding digital media health information seeking during pregnancy.**

| Theme | Example Quotes |
|---|---|
| **Strategies to manage physical and mental wellbeing during pregnancy** | "Yes. Mainly concerning symptoms and how other mums found relief" |
| | "Google things I've been told after appointments to double check what I was told" |
| | "[I searched for] recipes and exercises" |
| | "How to deal with the extra emotions and anxiety" |
| **Reassurance seeking** | "Yes. Searching for whether symptoms are normal for the stage of pregnancy I am at. . ." |
| | "Not really. Health (sic) I would usually ask the doctor. I only google to find out if something is normal" |
| | "In regards to my chronic illness and new medicine to treat it and its affects on pregnancy" |
| | "Have been researching possible reasons for bleeding during pregnancy" |

**Table 4. Key themes in response to using digital media to track development of fetus and the percentage participants endorsed each theme.**

| Theme | Percentage |
|---|---|
| Track size and growth of fetus | 70.83 |
| Prenatal health, i.e. diet and exercise | 14.58 |
| Symptoms to expect | 10.42 |
| Not specified | 4.17 |

## Digital media use and MFA

A one-way between groups ANOVA was conducted to determine if MFA differed between three groups of digital media use for social and emotional support. Participants were divided into three groups based on their amount of digital media use for social/emotional support: less than once a week, ($M = 90.8$, $SD = 3.85$, $n = 14$), between one and five times a week ($M = 88.5$, $SD = 2.85$, $n = 17$), and daily use ($M = 87.5$, $SD = 3.44$, $n = 17$). The assumption of normality was violated for the less than once a week group ($p = .047$), however, the test is considered robust to violations given the sample size. There was no evidence to suggest that MFA scores differ as a function of usage frequency for social and emotional support, $p = .788$.

The size and direction of the relationship between the overall score of digital media usage and MFAS were analysed. The Pearson's correlation was positive but non-significant, $r(46) = .144$, $p = .330$, indicating limited evidence to suggest a relationship between MFA scores and overall digital media use for information or support (see Fig 2).

## Digital media use and psychological wellbeing

To determine whether a relationship existed between digital media use, negative affect, social QOL, and self-criticism, separate correlation analyses were performed. Regarding negative affect, DASS-21 scores were ranked to conduct a Spearman's Rho test of correlation due to normality violations ($p < .01$). Results indicated the presence of a significant positive correlation between ranked DASS-21 scores and digital media usage, $r_s = .415$, $p = .003$, two-tailed, $n = 48$. A Spearman's Rho test of correlation indicated a negative relationship between ranked social QOL scores and the score of digital media usage, $r_s = .39$, $p = .007$, two-tailed, $n = 48$. A

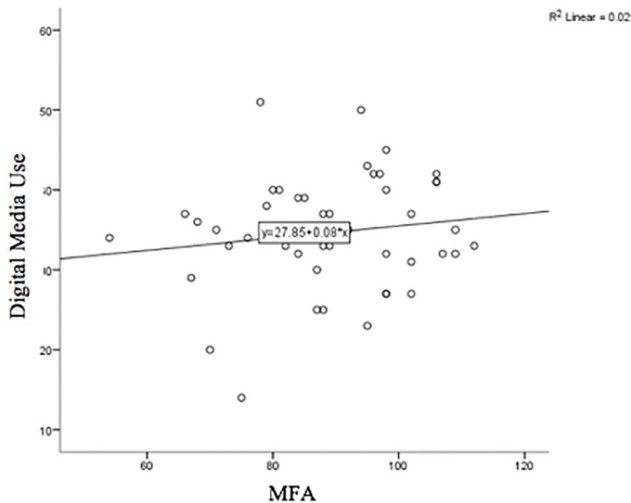

**Fig 2. Linear relationship between overall digital media usage for information and support during pregnancy and MFA scores.**

Pearson's correlation indicated a positive relationship between self-criticism (DEQ-SC-6 scores) and digital media use, $r(46) = .50$, $p < .001$. (see Fig 3)

## Discussion

The aims of the current study were to extend understanding of digital media use by women during pregnancy, and to determine whether there was a relationship between usage, psychological wellbeing and the developing emotional connection to the fetus. In our online survey, pregnant women reported using digital media for social and emotional support and obtaining health information for several reasons including determining what was normal about their pregnancy symptoms, understanding the development of their fetus, and learning what to

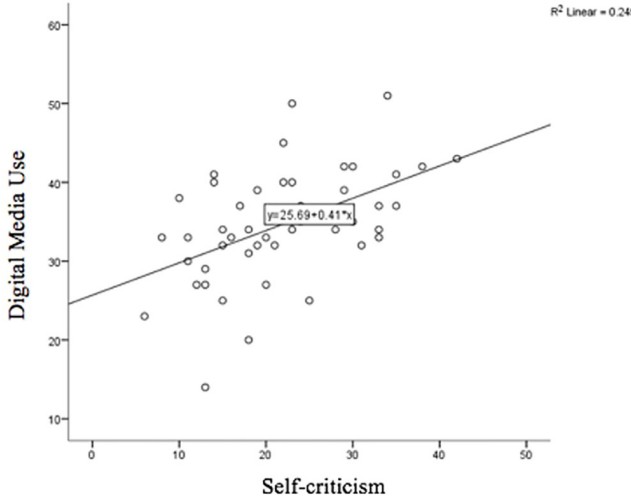

**Fig 3. Linear relationship between overall digital media usage for information and support during pregnancy and self-criticism scores.**

expect in their pregnancy journey. These findings are similar to the themes that have emerged from studies using focus groups [11, 12] and individual interviews [20] with pregnant women. The amount of digital media use was negatively correlated with age, a finding which is in line with a recent nationally representative survey of digital media usage where only 1.8% of 18 to 34 year olds stated that they did not use any form of digital media, compared to 7.8% of 35 to 49 year olds [8]. These findings highlight the importance of considering the contribution of digital media sources, alongside other traditional face-to-face programmes, when developing information and support for younger pregnant women.

Although obtaining social support is a commonly reported reason for using digital media [11, 13, 17, 18], which could serve to be protective against mental health issues, digital media use in this study was associated with higher self-criticism, higher negative affect and lower social quality of life. It is possible that women with higher negative affect increase their digital media usage in order to gain emotional and/or social support to help them through their experience of depression, anxiety or stress. Scherr and Brunet [46] in a study with younger Facebook users (74% female) found that the time spent on Facebook correlated with depression, but was mediated by the motives of distraction and relationship formation. These findings may suggest that individuals potentially use social media to distract themselves from their depressive symptoms, and to develop relationships to enhance social support. Considering the increased prevalence of depression and anxiety during pregnancy [47] this could explain why those who indicated higher digital media usage for information and support had higher negative affect. An alternate explanation is that higher digital media usage itself may be leading to higher levels of negative affect. Given that the internet provides opportunities to confidentially research and discuss sensitive topics and to be part of a community of people experiencing a similar life transition [19], further research is needed to understand why for some individuals online interactions may have a negative impact on their psychological wellbeing. Past research has found associations between higher social media site visits per week and increased odds of depression in a sample of 1787 adults [48]. A recent analysis of the feedback posted on online message boards has also highlighted the potential negative effect of information-seeking online during pregnancy. While Denton et al. [49] reported that women are receiving social support and information pertaining to medication safety from online forums, they are often subjected to judgment by other online users and report confusion over the differing information they found. Mitchell and Hussain [50] also found that problematic (dependent and addictive) smartphone use was positively related to excessive reassurance seeking. This trait of excessive reassurance seeking should be further examined within the perinatal period to determine whether it could explain the positive association between digital media use and negative affect found in the current study.

Digital media presents limitless opportunities to make comparisons between the self and how others are portrayed. For individuals who place high levels of expectations on themselves during pregnancy, rates of digital media use may reflect a need to validate their own behaviours and reassurance-seek [11]. Engaging in these comparison behaviours may lead vulnerable individuals to feelings of guilt, shame and overall self-criticism [2]. This may explain the present study's finding of digital media use being negatively correlated with self-criticism. Further research needs to examine these variables together to determine the nature of the reciprocal relationship between digital media use and self-criticism, especially given that recent research has suggested women's levels of self-criticism increase postnatally [51]. It is important to better understand whether digital media use is leading to increased levels of self-criticism during pregnancy, or if women who are self-critical are more likely to use digital media to reassure them than those who do not have high levels of self-criticism. Understanding the direction of the effect will provide opportunities for more tailored support for those engaging with digital media during pregnancy.

Despite past research indicating that women in the transition to parenting for the first time gained support through hearing or reading stories of other women online [20], our results indicated an inverse relationship where those engaging with higher use had lower social QOL. Vachkova, Jezek, Mares and Moravcova [52] found a clear downward trend of social QOL across all three trimesters and, as such, it is possible that those women who are experiencing poorer social QOL are relying more on digital media for emotional and social support than those who have better social QOL. This may explain why women in the current study who have higher social QOL have lower digital media usage as they may not require as much emotional and social support through digital media. However, this also means that accessing social and emotional support through digital media is not associated with increased social QOL and that it may not act as an appropriate substitute for face-to-face support from partners, friends and family. Considering the increasing digital nature of society, an important research direction will be to better understand how to best utilise face-to-face and digital interactions to support social QOL for women during pregnancy.

Although MFA is proposed to be driven by a mother's desire to know, protect and care for her child [27], and women in the current study reporting using the internet to inform themselves about the development of their fetus, we found no evidence that digital media use was associated with MFA. It is possible that that the gathering of information and support online for pregnancy experiences is an individual-focused experience, rather than a behavior which reflects a woman's developing "affiliation and interaction" [28 p.282] with her fetus. However, it is important to note that the current study had no minimum gestational age for participation. Previous research suggests that MFA begins developing from around 10 weeks gestation [53] and typically strengthens throughout pregnancy [54] as fetal movements increase [55]. Items on the MFA scale such as "I enjoy watching my tummy jiggle as the baby kicks inside" and "I poke my baby to get him/her to poke back" lack relevance to early pregnancy. An inclusion criteria of at least 20-weeks gestation or in the third trimester of pregnancy as used in previous research [27, 55, 56] would be important in developing further research on MFA and digital media use. The development of a revised MFAS for earlier stages in pregnancy is a potential next step for furthering our understanding of maternal psychological wellbeing throughout the pregnancy journey and the emergence of the attachment relationship.

This study has several strengths and limitations. A strength of the study is the heterogeneity of the sample, with the educational and cultural background reflective of the community the sample is drawn from. A limitation is that these findings may not be reflective of the experiences of women with high risk pregnancies, who may have very different information needs. Future studies should examine pregnancy risk profiles. Previous studies have used focus groups and interviews with small numbers of participants [11, 12, 21] or discourse analysis of mothering boards [19], while the current study's methodology allowed for a larger sample size and included validated self-report measures of psychological wellbeing and MFA. Other limitations include participant self-selection and the potential for response biases as a result of using online recruitment and survey methodology. However the purpose of the study was to examine when and how expectant mothers use digital media, not whether they are using digital media or not. An online approach further aligns with the high prevalence of digital media use in society, and increasing evidence that online research opportunities provides access to population groups that may be harder to reach in person (e.g. [57]) such as women at all stages of pregnancy. This study also had a high attrition rate, which may limit the generalisability of the data: approximately half of those who consented to the study, failed to complete the first survey. It is possible that a proportion of those participants who dropped out are those who use digital media regularly, i.e. interacting on Facebook, but do not use it for information and support during pregnancy. Advertising on Facebook may also have attracted women who ultimately did not have the time to complete an entire

survey, whereas other digital media forms such as mothering boards may be a more successful mode of recruiting participants who were willing to spend time completing in depth usage surveys (e.g. [19]). Further research should consider multiple online forums for recruitment. A further limitation is that the developed digital media questionnaire, although informed from questions used in other studies [13, 36], was not validated by consumers.

## Conclusions

Digital media use provides opportunities for individuals to seek out information and support, as and when it is wanted and needed. Despite evidence that pregnant women are using digital media to gain information and emotional and social support, outcome measures suggest that higher use of digital media can have a potentially negative effect on psychological wellbeing. In the current study, higher digital media use was correlated with higher self-criticism, lower quality of life, and high negative affect. The direction of these relationships requires further investigation. Increasingly, attention is being drawn to the need for the websites of governments and leading industry providers to contain pregnancy health-related information (e.g., about nutrition, sleep, physical activity) that is up-to-date with current research-led guidelines [14, 58]. Further research is needed to better understand how internet sources inform, support, or potentially challenge, the psychological wellbeing of individuals during pregnancy (see also [49]). It is crucial that all women have access to high-quality research-led information and the psychological support they need during pregnancy, irrespective of whether they choose to obtain that information through in-person interactions or through digital interactions.

## Supporting information

**S1 Appendix. Digital media questionnaire.**
(DOCX)

**S1 Dataset. Dataset.**
(XLSX)

## Acknowledgments

Thank you to Elise Kunkler for contributing to the data collection, and to the women who participated in this study. We acknowledge the traditional custodians of the land on which this research was conducted.

## Author Contributions

**Conceptualization:** Michelle L. Townsend, Jane S. Herbert.

**Data curation:** Melissa Smith, Annaleise S. Mitchell, Jane S. Herbert.

**Formal analysis:** Melissa Smith, Annaleise S. Mitchell.

**Funding acquisition:** Jane S. Herbert.

**Investigation:** Melissa Smith.

**Methodology:** Michelle L. Townsend, Jane S. Herbert.

**Project administration:** Melissa Smith.

**Resources:** Jane S. Herbert.

**Supervision:** Michelle L. Townsend, Jane S. Herbert.

**Validation:** Annaleise S. Mitchell.

**Visualization:** Melissa Smith, Annaleise S. Mitchell.

**Writing – original draft:** Melissa Smith, Jane S. Herbert.

**Writing – review & editing:** Melissa Smith, Annaleise S. Mitchell, Michelle L. Townsend, Jane S. Herbert.

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
