## [Editor Report · Decision Letter 0]

11 Mar 2020

PONE-D-20-05764

The relationship between digital media use during pregnancy, maternal psychological wellbeing, and maternal-fetal attachment

PLOS ONE

Dear A/Prof Herbert,

Thank you for submitting your manuscript to PLOS ONE. After careful consideration, we feel that it has merit but does not fully meet PLOS ONE’s publication criteria as it currently stands. Therefore, we invite you to submit a revised version of the manuscript that addresses the points raised during the review process.

This study describes how women were asked about their use of digital media to assist them in knowledge, and reassurance-seeking during pregnancy. As such, it is a relevant and contemporary topic for maternity care. This manuscript has potential; however, it needs revision before I would send it for peer review. I suspect that much of my comment here relates to transitioning from a thesis to a journal publication. I would be happy to see a revision that covers the following. Future peer-review may identify other areas to be addressed.

General comments

Although PLOS One does not specify this, I would encourage you to avoid the term “statistically significant” and replace it with commentary like “there was limited evidence to support…” or “we found no evidence of a difference …” See https://community.cochrane.org/sites/default/files/uploads/inline-files/Interpreting%20statistical%20significance.pdf

For example, in line 14 of the Abstract, you could write: “There was no evidence of a relationship between digital media use and maternal-fetal attachment” (and give data as evidence, see below)

Referencing in text: The Vancouver style, as I understand it, only imports the reference number in sequence as you have mostly done, rather than listing more than one author. For example, line 45, “Akiki, Avison [13] found …”  needs to read as “Akiki et al [13] found …”.

ABSTRACT

The survey is noted, however, no detail is provided about how findings were analysed, eg, mean, SD, assigning “formulated meanings” to comments, etc. No actual results are provided, only a comment about no association.

Measures, lines 131-134: I don’t understand how the questions used could result in free text responses. They appear to indicate the need for a “yes/no” response. How, then, did you ask women to provide free text?

RESULTS

Table 1 is usually the summary of participants’ characteristics. You could then reduce the commentary of these in text.

The current Table 1 only contains a small amount of data, which could be combined with the data in Table 3. Indeed, much of the text could be converted to table here, also incorporating the linear relationship analyses – the latter are currently figures which would not add anything to a journal publication and need to be removed. The detail in Table 2 is interesting – however, it would be helpful to have more detail in the methods section about what “formulated meaning” is, how many people go through the comments to determine this and how consensus on these meanings is achieved. You could provide a second quote for each of the three areas here. Also, there could be the potential to provide some quotes that did not support the “formulated meanings” and how you dealt with these.

In reporting mean and standard deviation, please use one decimal point for the mean and two for the SD. You can summarise the comments about not meeting normal distribution – you have noted that the test used was robust given the sample size. For the ANOVA results, you only need the p-value, rather than all the detail.

Discussion

This is good overall.

Line 320, the word “psychology” should be “psychological”.

Line 381, the word “reflects” would be better as “aligns with” as you initiated this method, rather than it reflecting the mode of survey that participants responded to when provided with a choice of mode.

We would appreciate receiving your revised manuscript by Apr 25 2020 11:59PM. To enhance the reproducibility of your results, we recommend that if applicable you deposit your laboratory protocols in protocols.io, where a protocol can be assigned its own identifier (DOI) such that it can be cited independently in the future. For instructions see: http://journals.plos.org/plosone/s/submission-guidelines#loc-laboratory-protocols

We look forward to receiving your revised manuscript.

Kind regards,

Christine E East

Academic Editor

PLOS ONE

Journal Requirements:

2) Please include additional information regarding the survey or questionnaire used in the study and ensure that you have provided sufficient details that others could replicate the analyses. For instance, if you developed a questionnaire as part of this study and it is not under a copyright more restrictive than CC-BY, please include a copy, in both the original language and English, as Supporting Information.

3) Please ensure that you include your title page at the beginning of your main document and remove the individual 'title page' file.
---

## [Author Response · Author response to Decision Letter 0]

24 Apr 2020

Dear Christine E East,

We appreciate your feedback and opportunity to revise our manuscript for submission at PLOS ONE. We have responded to each comment below, with line number references given to relevant changes in the attached ‘Revised Manuscript with Tracked Changes’ document. 

General comments:

• Although PLOS One does not specify this, I would encourage you to avoid the term “statistically significant” and replace it with commentary like “there was limited evidence to support…” or “we found no evidence of a difference …” See https://community.cochrane.org/sites/default/files/uploads/inline-files/Interpreting%20statistical%20significance.pdf

• For example, in line 14 of the Abstract, you could write: “There was no evidence of a relationship between digital media use and maternal-fetal attachment” (and give data as evidence, see below)

• Referencing in text: The Vancouver style, as I understand it, only imports the reference number in sequence as you have mostly done, rather than listing more than one author. For example, line 45, “Akiki, Avison [13] found …” needs to read as “Akiki et al [13] found …”

Authors’ response:

We have made your suggested changes to the language used when reporting statistical significance (examples of such changes on lines 307 – 310; 322). We have also updated the Vancouver referencing style in the manuscript to suit the journal’s referencing requirements. 

Abstract comments:

• The survey is noted, however, no detail is provided about how findings were analysed, eg, mean, SD, assigning “formulated meanings” to comments, etc. No actual results are provided, only a comment about no association.

• Measures, lines 131-134: I don’t understand how the questions used could result in free text responses. They appear to indicate the need for a “yes/no” response. How, then, did you ask women to provide free text?

Authors’ response:

We have included further details about the key findings in the abstract, including means, standard deviations, and significance values. We have elaborated in the measures section of the methods, to clarify that the questionnaire had the option of providing free text qualitative response (see lines 147 – 150). The digital media use questionnaire is now included as supporting information.

Results comments:

• Table 1 is usually the summary of participants’ characteristics. You could then reduce the commentary of these in text.

• The current Table 1 only contains a small amount of data, which could be combined with the data in Table 3. Indeed, much of the text could be converted to table here, also incorporating the linear relationship analyses – the latter are currently figures which would not add anything to a journal publication and need to be removed. The detail in Table 2 is interesting – however, it would be helpful to have more detail in the methods section about what “formulated meaning” is, how many people go through the comments to determine this and how consensus on these meanings is achieved. You could provide a second quote for each of the three areas here. Also, there could be the potential to provide some quotes that did not support the “formulated meanings” and how you dealt with these.

• In reporting mean and standard deviation, please use one decimal point for the mean and two for the SD. You can summarise the comments about not meeting normal distribution – you have noted that the test used was robust given the sample size. For the ANOVA results, you only need the p-value, rather than all the detail.

Authors’ response:

Thank you for your recommendations to the results section. We have taken your feedback on adjusting mean and SD decimal points and re-considered how we presented information in-text versus table form. As there were several variables in participant characteristics that we believe would have been difficult to present in table form clearly we decided to leave Table 1 as is and the participant characteristics in written form,

We agree that the term formulated meanings was a poor choice. We have revised this section to explain how the thematic analysis was conducted in line with Braun and Clarke’s (2006) procedure. Further details are now provided in the methods data analysis section of the manuscript (see lines 220-226). We also provided additional quotes in Table 2 of the results to illustrate the themes and associated meanings we identified. 

Discussion comments:

• This is good overall.

• Line 320, the word “psychology” should be “psychological”.

• Line 381, the word “reflects” would be better as “aligns with” as you initiated this method, rather than it reflecting the mode of survey that participants responded to when provided with a choice of mode.

Authors’ response:

We have revised these grammatical errors.

Kind regards,

Corresponding Author, Associate Professor Jane Herbert

---

## [Decision Letter · Decision Letter 1]

14 Jul 2020

PONE-D-20-05764R1

The relationship between digital media use during pregnancy, maternal psychological wellbeing, and maternal-fetal attachment

PLOS ONE

Dear Dr. Herbert,

Thank you for submitting your manuscript to PLOS ONE. After careful consideration, we feel that it has merit but does not fully meet PLOS ONE’s publication criteria as it currently stands. Therefore, we invite you to submit a revised version of the manuscript that addresses the points raised during the review process.

The reviewers propose a number of areas to address, including the need to describe in greater detail the gaps in the research evidence relating to psychological wellbeing and digital media use during pregnancy and further description of the thematic analysis and contribution of the participants.

We look forward to receiving your revised manuscript.

Kind regards,

Christine E East

Academic Editor

PLOS ONE

Reviewers' comments:

Reviewer's Responses to Questions

**Comments to the Author**

1. If the authors have adequately addressed your comments raised in a previous round of review and you feel that this manuscript is now acceptable for publication, you may indicate that here to bypass the “Comments to the Author” section, enter your conflict of interest statement in the “Confidential to Editor” section, and submit your "Accept" recommendation.

Reviewer #1: All comments have been addressed

Reviewer #2: (No Response)

2. Is the manuscript technically sound, and do the data support the conclusions?

Reviewer #1: Partly

Reviewer #2: Partly

3. Has the statistical analysis been performed appropriately and rigorously? 

Reviewer #1: Yes

Reviewer #2: Yes

4. Have the authors made all data underlying the findings in their manuscript fully available?

Reviewer #1: Yes

Reviewer #2: Yes

5. Is the manuscript presented in an intelligible fashion and written in standard English?

Reviewer #1: No

Reviewer #2: Yes

6. Review Comments to the Author

Reviewer #1: Please see attached document for my comments.

Reviewer #2: This manuscript reports on the relationship between digital media use during pregnancy, psychological wellbeing and maternal-fetal attachment which is an important issue in contemporary maternity care and seeks to improve our understanding of the use and effect of digital media for pregnant women.

The strength of this paper is a clear focus on the experiences of women when engaging with digital media throughout pregnancy.

This paper has potential to provide valuable insight into the views and experiences of pregnant women in the use of digital media.

There are several opportunities to strengthen the quality of the manuscript.

Abstract:

Line 35 – The statement ‘There are a number of benefits and limitations of this medium for providing information and support for young women during pregnancy’ should be revised as this is not supported by the evidence in the paper. There is insufficient evidence to support the statement that only ‘young’ women use this medium.

Introduction:

Line 46 – You have made a statement about the diagnosis of perinatal depression. There is no relevance demonstrated between this statement and the study. Unless there is a connection made, I would suggest removing this statement.

Line 58 - Suspected typo (eg 12) as a reference. Please address.

Line 71 – “women reported gaining knowledge and support by sharing their own stories and hearing other women’s stories”. I would suggest this evidence supports the use of online formats for this interaction. This needs to be clarified otherwise it is assumed you are referring to face to face.

Measures:

Line 140 – Was this questionnaire validated by consumers? If so, this should be mentioned. If not, this should be addressed.

Line 178 – “The WHOQOL-Brief was developed by the [36]” – Reference to the World Health Organisation (WHO) should be standardised throughout the paper.

Results:

Table 2 Line 268 – This study lends itself to thematic analysis of the views of women. However, this table does not demonstrate clearly what the views of women were and what themes emerged. The theme “Information and Reassurance seeking” seems to also be supported by the comment “Searching for whether symptoms are normal for the stage of pregnancy I am at…” but it has been attributed to a different theme “Determining what is normal” which is also reassurance seeking. Reconsideration of the themes and the analysis in general is recommended as the examples used do not support the thematic analysis described.

The quote “Not really. Health I would usually ask the doctor. I only google to find out if something is normal” should have (sic) inserted after ‘Health’ as the sentence does not make sense without this.

Discussion:

Pregnancy risk profile should be mentioned here as a limitation to this study. Women who are considered high risk have a very different experience of pregnancy (including information seeking) than low risk women. This needs to be acknowledged as a limitation of this study because this has not been considered.

Overall this manuscript requires major revision before it could be accepted for publication particularly with respect to the thematic analysis and contribution of the participants.

7. PLOS authors have the option to publish the peer review history of their article (what does this mean?). If published, this will include your full peer review and any attached files.

Reviewer #1: **Yes: **Noushin Arefadib

Reviewer #2: **Yes: **Fiona Faulks

---

## [Author Response · Author response to Decision Letter 1]

13 Nov 2020

Thank you for the thoughtful comments and suggestions from yourself and your reviewers. We appreciate such helpful and timely feedback. We apologise for the long delay in our resubmission, which was due to the impact of the pandemic on our workforce. We thanks the reviewers for their time and valuable feedback on this manuscript. We believe we have addressed all issues raised by reviewers. We have responded to each of the points raised below, and substantially revised the presentation of our thematic analysis and the contribution of the participants as requested. We believe this has strengthened the paper conceptually and empirically, and hopefully better communicates our findings.

Reviewer 1 Background

You describe clearly the reasons why examining the relationship between digital media use and psychological wellbeing are really important. However, there isn’t a lot said about what is actually missing in the research evidence and why it’s important to address this specific gap. Please describe in greater detail the gaps in the research evidence relating to psychological wellbeing and digital media use during pregnancy. Does any existing research touch on this? (See for example Ginja et al., 2018. Associations between social support, mental wellbeing, self-efficacy and technology use in first-time antenatal women: data from the BaBBLeS cohort study). You do this rather clearly in lines 119-122, describing gaps in the research pertaining to MFA and digital media use. I recommend including something similar regarding psychological well-being and digital media use. 

Response: New text [line 61-62] has been added: Although only one study [15] has examined the impact of technology on maternal mental health in pregnancy, finding limited association, the authors argue the importance of understanding the type of technological activities undertaken to understand how they facilitate social support and wellbeing.

Reviewer 1 Grammar / spelling / structure- 

Response: All suggested revisions have been made as requested

Reviewer 1 Omitted references and information 

Line 79-80: “Despite these advantages, online behaviors also have the potential to reinforce anxiety, for example if the internet is being used as a means of excessive reassurance-seeking”. Please provide a reference for this sentence. 

Response: The text has been revised and reference added.

Line 81: Please provide context to the percentage figure presented by sharing the total participant numbers in the study you refer to. Or just say “most” without providing a percentage figure. 

Response: We have removed the percentage figure and have kept the statement most: “…found that most participants reported that their search for information was unrelated to antenatal appointments”.

Line 119-120: “Much of the current research on digital media use during pregnancy has focused on examining why pregnant women use digital media and what sources they use.” Please provide a reference for this sentence. Which studies have looked at this? 

Response: We have revised this text [line 130-132]:

“The literature reviewed above confirms the significant role that internet searches, online discussion forums, and apps, play in supporting the health information seeking and decision making of women during pregnancy [11,12,14,21,22,23].”

Line 131-133: Please provide additional information on how participants were recruited. For example: “A Facebook post describing the study was generated. This included a link to the study survey…”etc. 

Response: The revised text [line 143-148] now states:

“For the purposes of recruitment, a Facebook post describing the study was generated. This included a brief description and a direct link to study consent information and survey…. Recruitment occurred across a 3 month period from May to July, 2018.”

Line 133: please explain what you mean by ‘local’. Please name the specific location or just say Australia-wide. 

Response: The revised text [line 146] now states: 

“The study information was posted on various mothering, pregnancy support groups and community groups on Facebook, Australia-wide.”

Line 143-144: “Questions were informed by those used in focus groups and interviews in previous studies.” Please describe what these studies were about and provide a rationale for the decision to draw on these studies to develop questions for this particular study. 

Response: Further information has been provided on the rationale to use these studies [line 158-161]

“Questions were informed by the themes of information seeking and providing reassurance that have been identified in previous research conducted with Australian women examining experiences of using digital media for pregnancy and parenting purposes [11, 12, 13, 37].

Line 146-149: Was this a yes/no response option? Or free text? Please explain further. Describe the number of questions that allowed a qualitative response. 

Response: The description of the digital media questionnaire has been revised, and the number of questions and response type better identified. Lines 161-179.

Line 160: Please provide a reference for the DASS-21.

Response: The text has been revised and reference provided (line 184-5]

Line 178: “The WHOQOL-Brief was developed by the [36] as a shortened version of…” Please describe by who. Should this be ‘developed by the world health organization’? 

Response: This text has been revised [line 202-205]:

“The WHOQOL-Brief was developed by the World Health Organisation (WHO) [43] as a shortened version of the WHOQOL-100, measuring quality of life, applicable across cultures. The WHOQOL-Brief contains 26 questions across four domains: physical, psychological, social relationships and environment.” 

Line 178: “The WHOQOL-Brief was developed by the [36] as a shortened version of…” Please describe by who. Should this be ‘developed by the world health organization’? 

Line 194-195: Please explain why socio-economic data was collected and why it is not referred to anywhere else in the study. 

Response: The reference to socio-economic was an error and has been removed. The text has been revised [line 220-222]

“It consisted of questions regarding personal demographics (age, country of birth, marital status, and education levels) and pregnancy-related questions (current gestational age, current and previous pregnancy history).”

Line 329-330: “Although obtaining social support is a commonly reported reason for using digital media, which could serve to be protective against mental health issues…” Please provide a reference. 

Response: References have been added. 

Reviewer 1 comments - Results

Line 232-246: Please put the participant’s characteristics in a table. 

Response: A new table has been included in the manuscript outlining the participants demographics (Table 1).

Line 253: Was this participant able to share their specific reason? 

Response: The revised text is:“Only one participant indicated that none of the listed reasons were why they used digital media, and provided their own response of “having down time”. 

Line 247: Please remove “n =”. 

Response: Unfortunately we have not been able to identify where this is in our text.

Line 279: Please describe what the ‘other’ reasons were.

Response: The revised text is: Only two participants indicated ‘other’ reasons, and specified these as “reviews on doctor” and “just to get a general idea”

Reviewer 1 Comments Discussion

Line 334-336: Was the Scherr and Brune study about pregnant women? Please expand on this. 

Response: The study was not with pregnant women, it was with a general population sample with a mean age of 28.7 years.

The text now states [lines 359-361]: Scherr and Brunet [46] in a study with younger Facebook users (74% female) found that the time spent on Facebook correlated with depression, but was mediated by the motives of distraction and relationship formation."

Line 336-337: “This suggests that individuals may use social media to distract themselves from their depressive symptoms, and to develop relationships to enhance social support.” IS this a personal hypothesis or a finding from a study? Provide evidence for this statement. 

Response: This sentence has now been revised [line 361-363] to state:

“These findings may suggest that individuals potentially use social media to distract themselves from their depressive symptoms, and to develop relationships to enhance social support.”

Line 343-344: “…individuals for whom online interactions have a negative impact on their psychological wellbeing.” This statement is very definitive. Online interactions could potentially increase the likelihood of certain outcomes. Please revise this sentence. 

Response: The text now states [line 367-370]: 

“Given that the internet provides opportunities to confidentially research and discuss sensitive topics and to be part of a community of people experiencing a similar life transition [19], further research is needed to understand for some individuals online interactions may have a negative impact on their psychological wellbeing.”

Line 347: Is this study about potential negative psychological effects? Was it about pregnant women? Please explain this more clearly. 

Response: This study aimed to explore and further understand whether there was any association between the social, emotional and informational support pregnant women receive through digital media use and whether that is related to their psychological wellbeing. As this body of literature is relatively new and limited it is too early to determine whether there is a positive or negative association. 

Line 351: Please explain what is meant here by “problematic smartphone use”. 

Response: The text now states [line377-379]:

“Mitchell and Hussain [50] also found that problematic (dependent and addictive) smartphone use was positively related to excessive reassurance seeking.”

Line 356-359: Overall, the distinction between ‘social media’ and ‘digital media’ use becomes blurred throughout the discussion. This needs to be addressed, perhaps through a definition of what entails ‘digital media’ in this study. Were you looking specifically at social media? It’s not very clear. 

Response: Thank you for this feedback. We have now added in the introduction [lines 52-56] the following explanation:

“Digital media comprises of many forms

of digital technology that allow for the dispersal of information including; informative websites and applications on a smart phone [6]. A large component of digital media is social media, which refers to internet-based channels of mass communication enabling interactions among users, with the content being primarily user-generated [7].

We have also replaced instances of social media with digital media as appropriate throughout the manuscript. 

Line 363-364: Revise sentence. 

Response: This hanging sentence has now been revised. Line 396-398 in the revised manuscript

Line 370-371: Is the study referenced here about pregnant women? Please specify. 

Response: We have clarified the participants in this study [Line 396-398], who were women in the transition to parenting for the first time

Line 389-392: The study did, however, collect data on participants’ gestational age. Is it not possible to use this data to examine a relationship between MFA and gestational age? Why was data on gestational age gathered otherwise? 

Response: We only used information on Gestational Age in the current manuscript as a way to describe the range of pregnant women who took part in the study (along with whether this is a first or later pregnancy and a planned pregnancy). As we did not restrict participation to a particular gestational age range, our range is from 2 weeks to 39 weeks. Thus some women will have participated in the study prior to the emergence of MFA [from around 10 weeks [53] or after movements have been felt [55], and with it strengthen across time afterwards [54]. We therefore decided not to include an examination of the relationship between GA and MFA in the current manuscript given that it is a small dataset. We acknowledge in the limitation section [line 421-423] “An inclusion criteria of at least 20-weeks gestation or in the third trimester of pregnancy as used in previous research [27, 55, 56] would be important in developing further research on MFA and digital media use.”

Line 403: Please explain this limitation in greater detail. 

Response: We now refer directly to the issues of self-selection and potential responses biases as a result of the online data collection and survey methodology used. Line 434-435.

Consider the representativeness of your sample in your limitations. 

Response: New text added [line 426-430]

“A strength of the study is the heterogeneity of the sample, with the educational and cultural background reflective of the community the sample is drawn from. A limitation is that these findings may not be reflective of the experiences of women with high risk pregnancies, who may have very different information needs. Future studies should examine pregnancy risk profiles.”

Reviewer 1 comments - Conclusion

Line 422: Please replace “opposite effect” with “potentially negative effect”. 

Response: Replaced

Reviewer 2

Reviewer 2 comments Abstract

Line 35 – The statement ‘There are a number of benefits and limitations of this medium for providing information and support for young women during pregnancy’ should be revised as this is not supported by the evidence in the paper. There is insufficient evidence to support the statement that only ‘young’ women use this medium.

Response: The word 'young' has been removed.

Reviewer 1 Introduction

Line 46 – You have made a statement about the diagnosis of perinatal depression. There is no relevance demonstrated between this statement and the study. Unless there is a connection made, I would suggest removing this statement.

Response: This statement has been removed.

Line 58 - Suspected typo (eg 12) as a reference. Please address.

Response: Typo removed

Line 71 – “women reported gaining knowledge and support by sharing their own stories and hearing other women’s stories”. I would suggest this evidence supports the use of online formats for this interaction. This needs to be clarified otherwise it is assumed you are referring to face to face.

Response: This study description has been revised to clarify that these were online forums, and that the majority of women were not engaging actively in interaction. [line 77-82]

“In individual interviews conducted by Johnson [20], women reported gaining knowledge and support from reading other women’s stories in online mothering spaces. Although some women reported the usefulness of participating in online social interaction, by asking questions or commenting on other women’s posts, the majority of women reported benefiting from these online spaces while remaining anonymous and engaging only as an observer”

Reviewer 2 Measures

Line 140 – Was this questionnaire validated by consumers? If so, this should be mentioned. If not, this should be addressed.

Response: New text in the limitations [line 448-450]

“A further limitation is that the developed digital media questionnaire, although informed from questions used in other studies [13,36], was not validated by consumers.”

Line 178 – “The WHOQOL-Brief was developed by the [36]” – Reference to the World Health Organisation (WHO) should be standardised throughout the paper.

Response: This text has been revised [line 202-205]

“The WHOQOL-Brief was developed by the World Health Organisation (WHO) [43] as a shortened version of the WHOQOL-100, measuring quality of life, applicable across cultures. The WHOQOL-Brief contains 26 questions across four domains: physical, psychological, social relationships and environment.”

Reviewer 2 Comments: Results

Table 2 Line 268 – This study lends itself to thematic analysis of the views of women. However, this table does not demonstrate clearly what the views of women were and what themes emerged. The theme “Information and Reassurance seeking” seems to also be supported by the comment “Searching for whether symptoms are normal for the stage of pregnancy I am at…” but it has been attributed to a different theme “Determining what is normal” which is also reassurance seeking. Reconsideration of the themes and the analysis in general is recommended as the examples used do not support the thematic analysis described.

Response: We thank the reviewers for this advice and have reviewed the thematic analysis and the themes that emerged to more clearly demonstrate the findings. The themes that emerged are now explained as [Table 3, line 291]

Strategies to manage physical and mental wellbeing during pregnancy

And

Reassurance seeking

Additional indicative quotes have been added to the two themes so that the reader can understand more about the types of responses women provided. 

The quote “Not really. Health I would usually ask the doctor. I only google to find out if something is normal” should have (sic) inserted after ‘Health’ as the sentence does not make sense without this.

Response: Added (sic) as suggested

Reviewer 2 Comments - Discussion

Pregnancy risk profile should be mentioned here as a limitation to this study. Women who are considered high risk have a very different experience of pregnancy (including information seeking) than low risk women. This needs to be acknowledged as a limitation of this study because this has not been considered.

Response: We thank the reviewer for this suggestion. New text added to the limitations [line 428-430]

“A limitation is that these findings may not be reflective of the experiences of women with high risk pregnancies, who may have very different information needs. Future studies should examine pregnancy risk profiles.”

---

## [Editor Report · Decision Letter 2]

1 Dec 2020

The relationship between digital media use during pregnancy, maternal psychological wellbeing, and maternal-fetal attachment

PONE-D-20-05764R2

Dear Dr. Herbert,

We’re pleased to inform you that your manuscript has been judged scientifically suitable for publication and will be formally accepted for publication once it meets all outstanding technical requirements.

Kind regards,

Christine E East

Academic Editor

PLOS ONE
---

## [Editor Report · Acceptance letter]

3 Dec 2020

PONE-D-20-05764R2 

The relationship between digital media use during pregnancy, maternal psychological wellbeing, and maternal-fetal attachment 

Dear Dr. Herbert:

I'm pleased to inform you that your manuscript has been deemed suitable for publication in PLOS ONE. Congratulations! Your manuscript is now with our production department. 

Kind regards, 

on behalf of

Dr. Christine E East 

Academic Editor

PLOS ONE